

# Genome-wide identification and co-expression network analysis of *Aux/IAA* gene family in *Salvia miltiorrhiza*

Bin Huang, Yuxin Qi, Xueshuang Huang and Peng Yang

Hunan Provincial Key Laboratory for Synthetic Biology of Traditional Chinese Medicine, School of Pharmaceutical Sciences, Hunan University of Medicine, Huaihua, Hunan, China

## ABSTRACT

The auxin/indole-3-acetic acid (Aux/IAA) gene family serves as a principal group of genes responsible for modulating plant growth and development through the auxin signaling pathway. Despite the significance of this gene family, the identification and characterization of members within the well-known Chinese medicinal herb *Salvia miltiorrhiza* (*S. miltiorrhiza*) have not been thoroughly investigated. In this study, we employed bioinformatics methods to identify 23 Aux/IAA genes within the genome of *S. miltiorrhiza*. These genes were classified into typical IAA and atypical IAA based on their domain structure. Our analysis of the promoter regions revealed that the expression of these genes is regulated not only by auxins, but also by other hormones and environmental factors. Furthermore, we found that the expression patterns of these genes varied across various tissues of *S. miltiorrhiza*. While our initial hypothesis suggested that the primary function of these genes was the interaction between *SmIAA* and *ARF*, gene co-expression network analysis revealed that they are also influenced by various other transcription factors, such as *WRKY* and *ERF*. The findings establish a sturdy basis for future investigations into the function of the Aux/IAA gene family and exhibit promising prospects for enhancing the genetics of this medicinal flora and its associated species.

## INTRODUCTION

Auxin, the first identified plant hormone, is present ubiquitously in all plants and participates in every stage of plant growth and development (*Guilfoyle & Hagen, 2007*). Indole-3-acetic acid (IAA) is a common plant auxin (*Luo, Zhou & Zhang, 2018*; *Wu et al., 2017*) and plays a crucial role in many processes of plant growth and development, such as cell division, embryogenesis, morphogenesis, directional response, dormancy extension, apical dominance, and tissue differentiation (*Kepinski & Leyser, 2005*; *Leyser et al., 1996*; *Leyser, 2017*; *Luo, Zhou & Zhang, 2018*; *Mironova et al., 2017*).

The response and signal transduction of plant cells to auxin are primarily mediated through the TIR1/AFB-Aux/IAA-ARF signaling pathway in the nucleus (*Guilfoyle & Hagen, 2007*). In this signaling pathway, auxin signal transduction is primarily controlled by auxin/indole-3-acetic acid (Aux/IAA) and auxin response factor (ARF) (*Lavy & Estelle, 2016*). In an environment with low concentrations of auxin, ARF and Aux/IAA protein

Corresponding authors
Xueshuang Huang,
xueshuanghuang@126.com
Peng Yang,
yangpenghnum@163.com

form a heterodimer and bind to the TOPLESS (TPL) protein and histone deacetylase (HDACs) to inhibit the response of ARF to auxin (*Luo, Zhou & Zhang, 2018*; *Parry et al., 2009*; *Winkler et al., 2017*). However, when the concentration of auxin is high in the environment, auxin binds to the TIR1 receptor, and the resulting SCF$^{TIR1}$ complex breaks down the dimer formed by Aux/IAA protein and ARF protein through ubiquitin (*Szemenyei, Hannon & Long, 2008*; *Winkler et al., 2017*). At the same time, proteases degrade Aux/IAA protein to restore ARF's activity (*Luo, Zhou & Zhang, 2018*; *Parry et al., 2009*; *Winkler et al., 2017*).

Aux/IAA protein is a crucial gene involved in the early response to auxin, and plays a vital role in regulating plant growth and development (*Luo, Zhou & Zhang, 2018*). The protein typically consists of four distinct domains, as described in previous studies (*Dreher et al., 2006*; *Guilfoyle & Hagen, 2007*). Domain I, acting as a transcriptional repressor, can be repressed by the adjacent promoter region. It contains an EAR motif associated with the ethylene response factor (ERF) (*Szemenyei, Hannon & Long, 2008*; *Tiwari, Hagen & Guilfoyle, 2003*). The LxLxL motif in the EAR motif of domain I may attract TPLs, which, when combined with the conserved N-terminal TPD domain of TPLs, create a common repressor, thus serving as transcriptional suppressors of downstream auxin regulating genes. Domain II serves as the degradation determinant, featuring the conserved TIR1/AFB recognition sequence GWPPV, which plays a vital role in the stability of the Aux/IAA protein (*Shimizu-Mitao & Kakimoto, 2014*). The GWPPV motif is a combination of five amino acids, namely Gly-Trp-Pro-Pro-Val. Mutations in the GWPPV recognition sequence may delay the degradation of Aux/IAA and disrupt the auxin signal transduction pathway (*Shimizu-Mitao & Kakimoto, 2014*). Domain III, with its βαα-fold structure, bears structural and functional resemblance to the DNA recognition motif in Arc and Met J repressors (*Dreher et al., 2006*; *Liu et al., 2011*; *Luo, Zhou & Zhang, 2018*). The fourth and final domain, Domain IV, contains one acidic domain and one SV40 type NLS (PKKKRKV) domain (*Dreher et al., 2006*; *Liu et al., 2011*; *Luo, Zhou & Zhang, 2018*). Domains III and IV are identical to the C-terminal dimerization domain (CTD domain) of ARF proteins, capable of generating type I/II PB1 domains that interact with Aux/IAA and ARF family proteins to regulate ARF activity (*Guilfoyle & Hagen, 2007*; *Luo, Zhou & Zhang, 2018*; *Winkler et al., 2017*).

Currently, through genome-wide analysis, the Aux/IAA gene family has been identified in over 30 plant species. For example, *Arabidopsis thaliana* (*A. thaliana*) contains 29 IAA proteins (*Liscum & Reed, 2002*), *Oryza sativa* (*O. sativa*) contains 31 members (*Jain et al., 2006*), *Solanum tuberosum* (*S. tuberosum*) and *Vitis vinifera* (*V. vinifera*) all contain 26 members (*Çakir, Kiliçkaya & Olcay, 2013*; *Gao et al., 2016*). Certain Aux/IAA genes have been shown to play a crucial role in plant growth, among which the AtIAA7 protein may inhibit flowering time under short-day light by negatively regulating the expression of *GA20ox1* and *GA20ox2* genes (*Luo, Zhou & Zhang, 2018*; *Mai, Wang & Yang, 2011*). The AtIAA8 protein interacts with the AtARF6/8 protein and modulates the level of jasmonic acid (JA) to influence flower organ development (*Wang et al., 2013*). The AtIAA17 protein plays an important role in regulating leaf senescence (*Shi et al., 2015*). Functional analysis of the *AtIAA28* mutant revealed that *AtIAA28* promotes the

transcription of lateral root initiation genes in response to auxin signals (*Rogg, Lasswell & Bartel, 2001*).

*Salvia miltiorrhiza* (*S. miltiorrhiza*), a member of the genus *Salvia* within the *Labiatae* family, is well known for its dried roots and rhizomes, which serve as a traditional Chinese medicine (*Xu et al., 2016*). Due to its advantageous growth characteristics, including a simple growth environment, short reproductive cycle, small genome size, and ease of tissue culture, *S. miltiorrhiza* has been established as a model system for the study of medicinal plants (*Wang et al., 2021*; *Xu et al., 2016*). The Aux/IAA gene family plays a crucial role in regulating plant growth and development (*Guilfoyle & Hagen, 2007*), making the identification of these genes in *S. miltiorrhiza* of paramount importance for understanding the developmental process and cellular response of auxin in this medicinal plant. This information could have important implications for the development of new plant varieties with improved growth and stress tolerance, as well as for the utilization of *S. miltiorrhiza* as a valuable medicinal resource. However, in *S. miltiorrhiza*, the use of the whole genome to study the Aux/IAA gene family has not been reported.

In the present investigation, we executed a comprehensive analysis of the Aux/IAA gene family in *S. miltiorrhiza* through a genome-wide approach. As a result, we successfully identified a total of 23 members of the *S. miltiorrhiza* Aux/IAA gene family (designated as *SmIAA*). Subsequently, we performed bioinformatic analyses to evaluate the physicochemical properties and sequence features of these genes. In addition, we utilized RNA-seq data to investigate the tissue-specific expression profiles of the *SmIAAs*, and further established a co-expression network to uncover the co-expression relationships between the *SmIAAs* and *transcription factors*.

## MATERIALS AND METHODS

### Identification and characterization of the Aux/IAA gene family in *S. miltiorrhiza*

The Aux/IAA protein sequences of *A. thaliana* and *O. sativa* were downloaded from The Arabidopsis Information Resources (TAIR) database and Rice Genome Annotation Project (RGAP) database respectively. The genomic data of *S. miltiorrhiza* was downloaded from National Genomics Data Center (NGDC) database (project number GWHAOSJ00000000).

We employed two methods to identify potential members of the *SmIAAs*. First, we utilized the *A. thaliana* Aux/IAA protein sequence as a seed sequence for BLASTP homologous alignment in the *S. miltiorrhiza* local database. We then validated the domain structure of the resulting sequences using the Pfam database (*El-Gebali et al., 2018*) and retained only those sequences that contained the AUX_IAA domain as candidate members of the *SmIAAs*. Second, we utilized the iTAK database to comprehensively evaluate the entire *S. miltiorrhiza* protein sequence and identified additional members of the candidate *SmIAAs* (*Zheng et al., 2016*). By combining the candidate gene family members obtained by these two methods, we determined the final set of members comprising the *SmIAAs*.

To explore the evolutionary relationships among *SmIAAs*, we utilized the Neighbor-Joining (NJ) approach implemented in MEGA X software (*Kumar et al., 2018*), employing the following parameters: 50% partial deletion and 500 bootstrap replications. Additionally, we analyzed the basic physicochemical properties of the protein sequences using the Expasy online website (http://web.expasy.org/protparam/) and predicted their subcellular localization using the WoLF PSORT software (https://wolfpsort.hgc.jp/). To further investigate the chromosomal distribution of *SmIAAs*, we obtained relevant information from the *S. miltiorrhiza* genome annotation file and visualized it using the MG2C software (http://mg2c.iask.in/mg2c_v2.0/).

## Genetic analysis of Aux/IAA gene sequence and promoter cis-acting elements in *S. miltiorrhiza*

The structure of *SmIAA* gene sequence was analyzed by TBtools software according to the GFF3 file and CDS file of *S. miltiorrhiza* (*Chen et al., 2020*). The protein sequence domain of SmIAA proteins were mined by CD-search online database (https://www.ncbi.nlm.nih.gov/Structure/bwrpsb/bwrpsb.cgi). Finally, Tbtools software was used to visually analyze the results (*Chen et al., 2020*). Furthermore, GO enrichment analysis was performed using R software. In addition, the promoter sequence of 2,000 bp upstream of the start codon of the *SmIAAs* was extracted from the *S. miltiorrhiza* genome. Subsequently, the promoter cis-acting elements were analyzed using the online website PlantCARE (http://bioinformatics.psb.ugent.be/webtools/plantcare/html/), with the results of this analysis being visualized through TBtools software, as detailed in Table S1.

## Comparative analysis of Aux/IAA gene family across different species

In order to explore the genetic and evolutionary relationships among members of the Aux/IAA gene family in various species, a multiple sequence alignment of 23 SmIAA proteins, 29 AtIAA proteins, and 31 OsIAA proteins was performed using the online program Clustal Omega, followed by MEGAX to create the phylogenetic tree (Table S2), with the parameters used in line with previous analyses (*Kumar et al., 2018*; *Sievers & Higgins, 2014*). Furthermore, we utilized the Maximum Likelihood (ML) method to construct an additional evolutionary tree to compare with the results obtained using the NJ method (Fig. S4). This was done with the implementation of specific parameters, including a 50% partial deletion criterion and 500 bootstrap replications. The phylogenetic tree was shown using the online software Evolview (https://www.evolgenius.info/evolview/) (*Subramanian et al., 2019*). MEME, an online program (https://meme-suite.org/meme/), was used to assess conserved motifs.

## Collinearity analysis and selection pressure analysis

The GFF annotation file of the genome was utilized to acquire the chromosomal location data for *SmIAAs*. The MCScanX software, developed by *Wang et al. (2012)*, was subsequently employed to compute the collinear block and tandem repeat gene information. The TBtools software (*Chen et al., 2020*) was utilized to visualize the chromosomal distribution and collinear relationships of the *SmIAAs*. For each identified

homologous gene pair by MCScanX, the synonymous substitution rate (Ks) and non-synonymous substitution rate (Ka) were computed using KaKs_Calculator software (*Wang et al., 2010*) for each synonymous site. The Ka/Ks ratio was utilized to evaluate the rate of protein evolution and determine the selection pressure of the genes during evolution.

### RNA-seq analysis of *SmIAA* expression in *S. miltiorrhiza*

The RNA-seq data of *S. miltiorrhiza* were obtained from the NCBI database (PRJNA744957 & PRJNA798876). The received sequence read archive (SRA) files were first converted to the fastq format using the fastq-dump program. The downloaded raw data was then trimmed and preprocessed using TrimGalore (https://github.com/FelixKrueger/TrimGalore), and the resulting clean data was utilized for downstream analysis. To construct a genome index with default settings, the Hisat2-build command in Hisat2 (version 2.3.0) (*Zhang et al., 2021*) was used. The featureCounts software was then employed to calculate the raw counts of each gene in each sample. The standardized transcriptional abundance was determined by computing the transcripts per million (TPM) value. The raw counts of each sample were used to generate a gene expression matrix of 21 transcriptome samples (Table S3). Finally, the expression data for the *SmIAAs* was extracted and a corresponding heat map was generated using the TBtools software (*Chen et al., 2020*).

### Gene co-expression network analysis in *S. miltiorrhiza*

Transcription factors were identified in the entire protein sequence of *Salvia miltiorrhiza* using the iTAK online software developed by *Zheng et al. (2016)*. Gene co-expression network analysis was conducted using R software version 3.9.4 and the Weighted Gene Co-expression Network Analysis (WGCNA) package developed by *Langfelder & Horvath (2008)*. First, the goodSamplesGenes function within the WGCNA package was utilized to identify and filter out any genes with missing values. The soft threshold power of 14 was then selected to construct the weighted gene network, with parameter settings of minModuleSize = 30, mergeCutHeight = 0.25, and maxBlockSize = 6,000. Additionally, to ensure the biological significance of each co-expression module, GO enrichment analysis was performed for each module using R software.

### Gene expression analysis of *S. miltiorrhiza* using qPCR and transcriptomics data integration

Two-year-old *S. miltiorrhiza* seedlings were collected and frozen in a −80 °C refrigerator. The roots, stems, leaves, and flowers were used for specific expression detection. Fluorescent quantitative specific primers (Table S4) were designed using Primer 5.0, and *SmUBQ* was used as an internal reference gene. Total RNA was extracted using the RNAprep Pure Plant Plus Kit (Tiangen, Beijing) and reverse transcribed using the PrimeScriptTM reverse transcriptase kit (Takara Tokyo, Japan). Real-time PCR reactions were performed using the PerfectStart Green qPCR SuperMix Kit (Transgene, Beijing), and fluorescence signals were collected using a qTOWER 3G PCR instrument. Each

sample was analyzed in triplicate, and relative expression was calculated using the $2^{-\Delta\Delta CT}$ method (Table S5). The correlation between transcriptome data and qPCR data was calculated and visualized by Sangerbox website (http://sangerbox.com/) (*Shen et al., 2022*).

## RESULT

### Identification of *Aux/IAA* gene family members in *S. miltiorrhiza*

In this study, two distinct methods were employed to identify the Aux/IAA gene family across the entire genome of *S. miltiorrhiza*. The initial approach involved using *Arabidopsis* Aux/IAA protein sequences to conduct a BLASTp search on the proteomic data of *S. miltiorrhiza*. Subsequently, the results were subjected to domain detection *via* submission to the Pfam database (*El-Gebali et al., 2018*), and the undesired outcomes were removed. This led to the identification of a total of 23 potential Aux/IAA gene family members. The second method involved employing the iTAK database to analyze the complete protein sequences of *S. miltiorrhiza*, which also resulted in the identification of 23 candidate genes. By comparing the outcomes of the two methods, it was established that the *S. miltiorrhiza* genome contained 23 members of the Aux/IAA gene family. In order to simplify future research, the nomenclature of *Arabidopsis genes* was adopted, and the genes were labeled as SmIAA1-SmIAA23 based on their distribution across the evolutionary tree (Fig. S1).

A comprehensive analysis was conducted to investigate the physicochemical properties of the SmIAA protein sequence. The results revealed that the number of amino acids in the SmIAA protein ranged from 87 to 371 aa, with an average of 211.61 aa. The molecular weight of the SmIAA protein was found to be between 10,364.81 and 40,197.45 Da, while the theoretical isoelectric point ranged from 4.82 to 9.81. Moreover, the subcellular localization analysis indicated that some proteins, namely SmIAA1, SmIAA4, SmIAA5, SmIAA8, SmIAA10, SmIAA11, SmIAA14, SmIAA16, SmIAA18, SmIAA22, and SmIAA23, were located in the cytoplasm, whereas the remaining proteins were all localized in the nucleus (Table 1).

Furthermore, we referred to the genome annotation information of *S. miltiorrhiza* to identify the distribution patterns of *SmIAAs*, with the exception of *SmIAA23* on contig685. Our analysis revealed that these genes were situated on chromosomes other than chromosome 7, with most genes located at the upper and lower ends of the chromosome. The highest frequency of gene distribution was observed on chromosome 5 (eight genes), while the lowest was on chromosomes 6 and 8 (one gene). Moreover, the genes on chromosomes 2 and 5 were more densely distributed compared to those on other chromosomes, which were more sparsely and randomly distributed (Fig. S2).

### Analysis of gene structure and protein sequence

Previous studies have demonstrated that gene structural diversity is a significant driver for the evolution of many gene families (*Singh & Jain, 2015*). To further explore the gene structure of *SmIAAs* (Fig. 1A), we conducted an analysis which revealed variable splicing in all genes. The number of introns ranged from 1 to 6. Notably, some genes did not have

**Table 1 The physiological and biochemical features of *SmIAAs*.** The Aux/IAA gene family in *S. miltiorrhiza*: the number of members, molecular weight, isoelectric point, and the presence of conserved domains.

| Name | Gene ID | Accession ID | Chromosome localization | | | Mw (Da) | pI | CDS (bp) | Length (aa) | Subcellular localization | Homologs in Arabidopsis |
|------|---------|--------------|-------------------------|---|---|---------|-----|----------|-------------|--------------------------|-------------------------|
| SmIAA1 | EVM0000296.1 | GWHTAOSJ027615 | Chr8 | 5920653 | 5921878 | 21688.8 | 9.81 | 573 | 190 | Cytoplasm | AT3G62100 |
| SmIAA2 | EVM0009395.1 | GWHTAOSJ003786 | Chr1 | 57848442 | 57849325 | 20565.38 | 8.58 | 540 | 179 | Nucleus | AT1G04550 |
| SmIAA3 | EVM0004349.1 | GWHTAOSJ017483 | Chr5 | 872484 | 876005 | 24827.71 | 5.94 | 657 | 218 | Nucleus | AT3G62100 |
| SmIAA4 | EVM0022630.1 | GWHTAOSJ006473 | Chr2 | 327881 | 328425 | 10364.81 | 4.82 | 264 | 87 | Cytoplasm | AT1G04550 |
| SmIAA5 | EVM0009745.1 | GWHTAOSJ005170 | Chr1 | 73323215 | 73324050 | 16049.36 | 9.25 | 420 | 139 | Cytoplasm | AT5G57420 |
| SmIAA6 | EVM0021654.1 | GWHTAOSJ023551 | Chr6 | 60358018 | 60360065 | 20499.12 | 5.16 | 537 | 178 | Nucleus | AT1G04550 |
| SmIAA7 | EVM0001033.1 | GWHTAOSJ007657 | Chr2 | 13722326 | 13725721 | 40197.45 | 9.38 | 1116 | 371 | Nucleus | AT2G46990 |
| SmIAA8 | EVM0026407.1 | GWHTAOSJ019954 | Chr5 | 58348218 | 58351945 | 28501.2 | 8.64 | 810 | 269 | Cytoplasm | AT2G33310 |
| SmIAA9 | EVM0002334.1 | GWHTAOSJ016330 | Chr4 | 66298635 | 66301163 | 34131.41 | 9 | 927 | 308 | Nucleus | AT5G25890 |
| SmIAA10 | EVM0010171.1 | GWHTAOSJ019616 | Chr5 | 55028101 | 55031070 | 18580.66 | 8.17 | 498 | 165 | Cytoplasm | AT1G15580 |
| SmIAA11 | EVM0027652.1 | GWHTAOSJ019676 | Chr5 | 55711914 | 55716112 | 19848.71 | 5.41 | 534 | 177 | Cytoplasm | AT1G15580 |
| SmIAA12 | EVM0010443.1 | GWHTAOSJ015659 | Chr4 | 57556179 | 57562422 | 38312.09 | 7.6 | 1077 | 358 | Nucleus | AT3G23050 |
| SmIAA13 | EVM0025242.1 | GWHTAOSJ011536 | Chr3 | 21196572 | 21199823 | 36647.47 | 8.6 | 1023 | 340 | Nucleus | AT3G04730 |
| SmIAA14 | EVM0003584.1 | GWHTAOSJ019677 | Chr5 | 55738962 | 55740433 | 21585.91 | 8.15 | 594 | 197 | Cytoplasm | AT1G80390 |
| SmIAA15 | EVM0019034.1 | GWHTAOSJ019617 | Chr5 | 55050465 | 55052967 | 20718.73 | 7.72 | 573 | 190 | Nucleus | AT5G43700 |
| SmIAA16 | EVM0009041.1 | GWHTAOSJ007792 | Chr2 | 15980426 | 15981800 | 20799.85 | 6.73 | 561 | 186 | Cytoplasm | AT4G14550 |
| SmIAA17 | EVM0020724.1 | GWHTAOSJ007565 | Chr2 | 12458073 | 12462291 | 26535.11 | 8.96 | 735 | 244 | Nucleus | AT3G04730 |
| SmIAA18 | EVM0011993.1 | GWHTAOSJ020429 | Chr5 | 62919943 | 62922717 | 24660.27 | 6.02 | 687 | 228 | Cytoplasm | AT4G14550 |
| SmIAA19 | EVM0021151.1 | GWHTAOSJ010857 | Chr3 | 12283059 | 12284558 | 25314.25 | 7.54 | 684 | 227 | Nucleus | AT4G14550 |
| SmIAA20 | EVM0010879.1 | GWHTAOSJ020426 | Chr5 | 62894169 | 62895673 | 17843.23 | 5.31 | 489 | 162 | Nucleus | AT5G43700 |
| SmIAA21 | EVM0004282.1 | GWHTAOSJ004659 | Chr1 | 68141757 | 68142829 | 20661.38 | 6.61 | 561 | 186 | Nucleus | AT5G43700 |
| SmIAA22 | EVM0006475.1 | GWHTAOSJ007793 | Chr2 | 16036879 | 16038173 | 19453.26 | 5.39 | 519 | 172 | Cytoplasm | AT5G43700 |
| SmIAA23 | EVM0007611.1 | GWHTAOSJ000893 | contig685 | 54663 | 55050 | 10916.64 | 5.09 | 291 | 96 | Cytoplasm | AT4G14560 |

UTRs. *SmIAA2, SmIAA3, SmIAA4, SmIAA6, SmIAA13, SmIAA17*, and *SmIAA23* contained only CDS and no UTR.

To gain more insights into the properties of SmIAA proteins, we performed domain analysis using the Pfam database. The results showed that all proteins contained Aux/IAA domains, although their placement and size varied (Fig. 1B). We also conducted multiple sequence alignment to evaluate the conserved amino acid sites in the SmIAA proteins. The domains of SmIAA proteins were further divided into Domain I (EAR-motif), Domain II (Degron motif), Domain III (Basic motif), and Domain IV (OPCA-like motif) (Figs. 2 and S3). Some SmIAA proteins were found to be atypical Aux/IAA proteins, lacking at least one conserved domain. This structural diversity may provide these proteins with diverse functions, although further research is required.

## Phylogenetic analysis of SmIAA proteins

To investigate the potential relationship between typical and atypical AUX/IAA proteins, we identified and selected the Aux/IAA proteins from two model organisms, *A. thaliana*

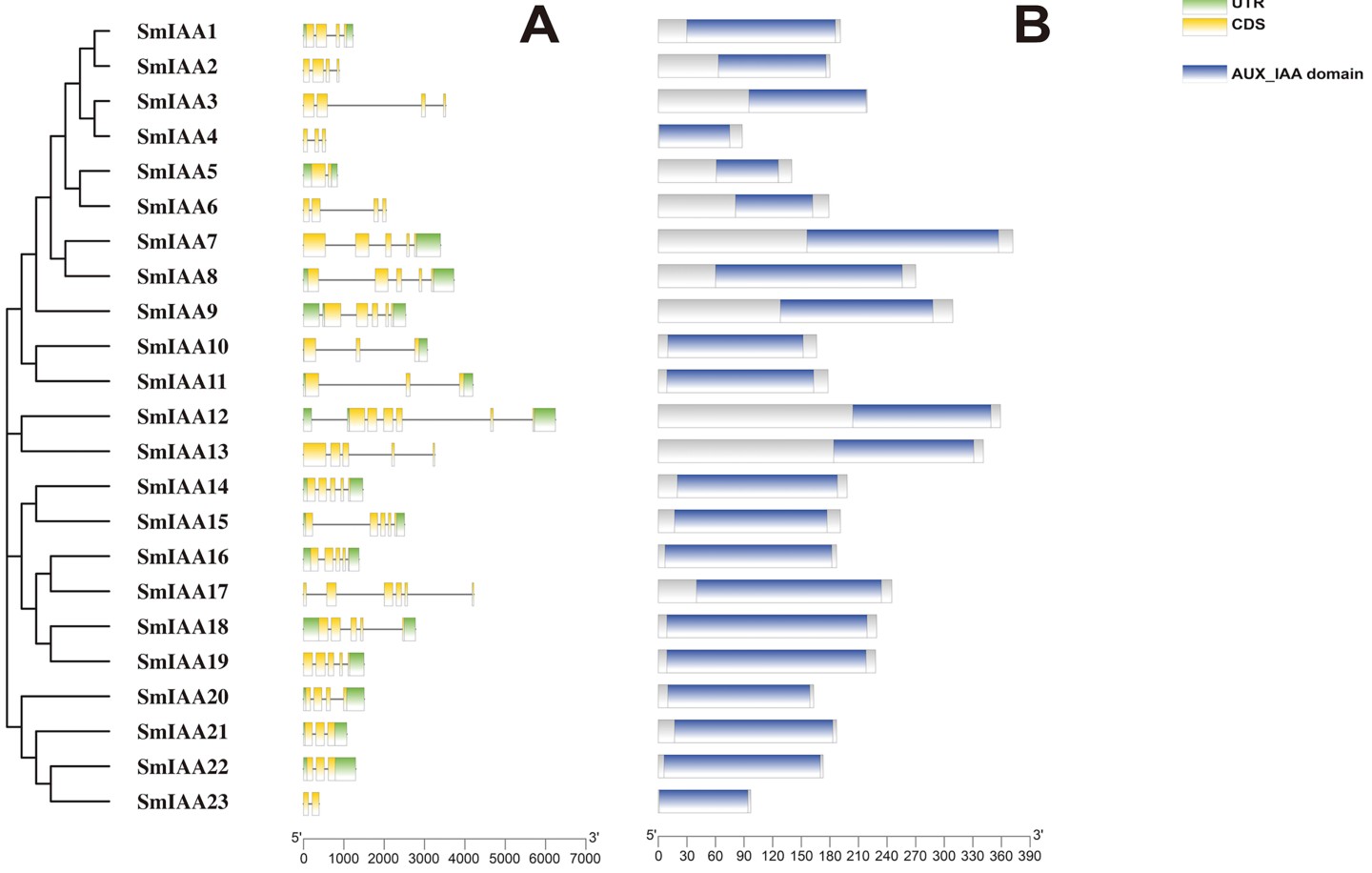

**Figure 1** **The gene structure and protein structure analysis of *SmIAAs*.** (A) The upper panel shows the gene structure of the *SmIAAs*. The CDS regions are represented by yellow boxes, respectively. The untranslated regions (UTRs) are represented by green boxes. The scale bar indicates the length of the gene structure in base pairs. (B) The protein structures of SmIAA proteins were predicted using the Pfam database. The AUX_IAA domain contained in SmIAA proteins is represented by a graded blue box. The evolutionary tree was constructed using the neighbor-joining (NJ) method, based on the multiple sequence alignment of SmIAA protein sequences. The evolutionary distances were calculated using the Poisson correction method. The whole figure was created using Tbtools, a software tool for the visualization and annotation of biological data.

and *O. sativa*, as well as *S. miltiorrhiza* for further study. We utilized MEGA X to construct phylogenetic trees of the Aux/IAA proteins in three species. The results obtained from the ML method and NJ method were largely consistent, indicating the reliability of our findings. Additionally, we employed the MEME database to analyze the motifs present in these proteins. The resulting evolutionary tree was primarily divided into two groups, I and II, respectively. As illustrated in Fig. 3, the evolutionary tree and conserved motifs were consistent with previous studies on the Aux/IAA family of plants. The majority of the members were classified in Group I, with nearly all of them being canonical Aux/IAA proteins possessing four canonical domains (motif1, 2, 3, and 4). On the other hand, nearly all the members of Group II were found to be atypical Aux/IAA proteins, lacking at least one conventional domain. Atypical Aux/IAA proteins have been found to be prevalent in the Aux/IAA gene family of various plants, where they play a significant role in plant

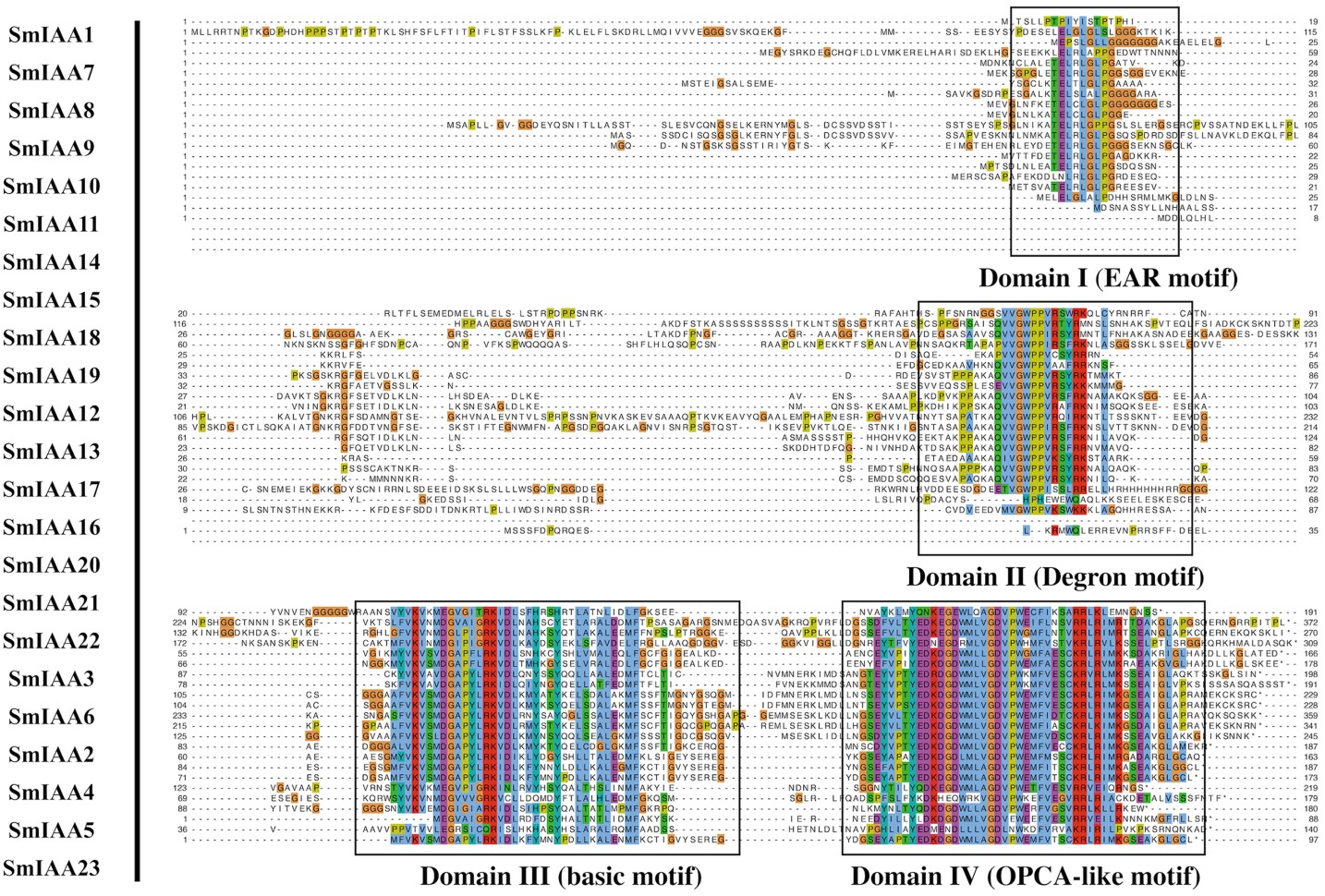

**Figure 2 Multiple sequence alignment analysis of SmIAA family proteins.** The figure shows the results of a multiple sequence alignment analysis of SmIAA proteins. The alignment was performed using the Clustal Omega program, and the results were visualized using the Jalview program. The different regions of the proteins are color-coded, with the conserved domains indicated by different colored boxes. The numbers above the alignment indicate the position of each residue in the protein sequence. The alignment reveals that the SmIAA proteins share a high degree of sequence conservation in the conserved domains, including the AUX/IAA domain and the domain II/III. The variable regions of the proteins are less conserved, reflecting their divergent functions. The alignment also reveals the presence of several conserved motifs, which may be involved in protein-protein interactions or other functions.

adaptability to changing environmental conditions, although their specific functions remain largely unknown (*Luo, Zhou & Zhang, 2018*).

## Analysis of collinearity and selection pressure

We conducted a collinear analysis of *SmIAAs* to elucidate their relationships further. The results showed that nine lineal homologous gene pairs were identified among all *SmIAAs*, suggesting that chromosomal replication processes play a crucial role in the amplification of the *S. miltiorrhiza* Aux/IAA gene family members (Fig. 4).

To gain a better understanding of the evolutionary processes of genes and proteins, studying selective interaction patterns can be a useful approach. Selection has been shown to be a valuable technique for speculating about gene function (*Wang et al., 2009*). The Ka/Ks value, which is the ratio of the two protein-coding genes' non-synonymous

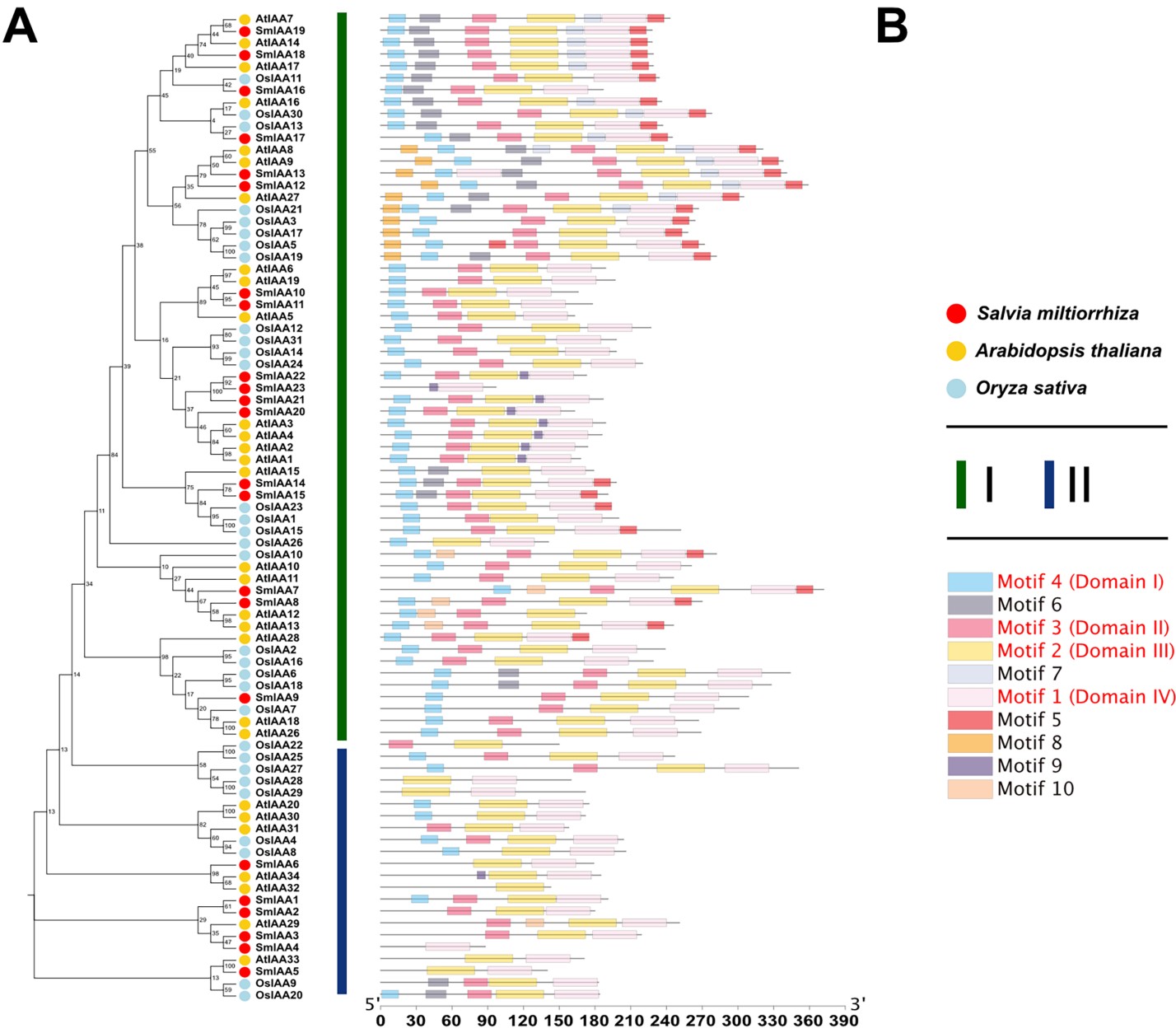

**Figure 3 Phylogenetic tree and conserved motif analysis of Aux/IAA family genes in *A. thaliana*, *O. sativa*, and *S. miltiorrhiza*.** (A) The evolutionary tree was constructed using the neighbor-joining (NJ) method, based on the multiple sequence alignment of Aux/IAA family protein sequences from *Arabidopsis thaliana*, *Oryza sativa*, and *Salvia miltiorrhiza*. The evolutionary distances were calculated using the Poisson correction method, and the numbers on the branches represent the bootstrap values (in percentage) from 500 replicates. The different clades are labeled with different colors, corresponding to the different species shown in the figure. All the sequences used to construct the evolutionary tree can be obtained from Table S2. (B) The conserved motifs of the Aux/IAA proteins were identified using the MEME Suite software. The different motifs are represented by different colored boxes. The motifs that are specially marked are related to the unique domain of Aux/IAA.

substitution rates (Ka) and synonymous substitution rates (Ks), can be used to identify whether the gene is under selection pressure. When Ka/Ks > 1, the gene is thought to be positively selected; when Ka/Ks = 1, the gene is thought to be purified and selected; and when Ka/Ks < 1, the gene is thought to be neutrally selected. Except for *SmIAA8/SmIAA14*

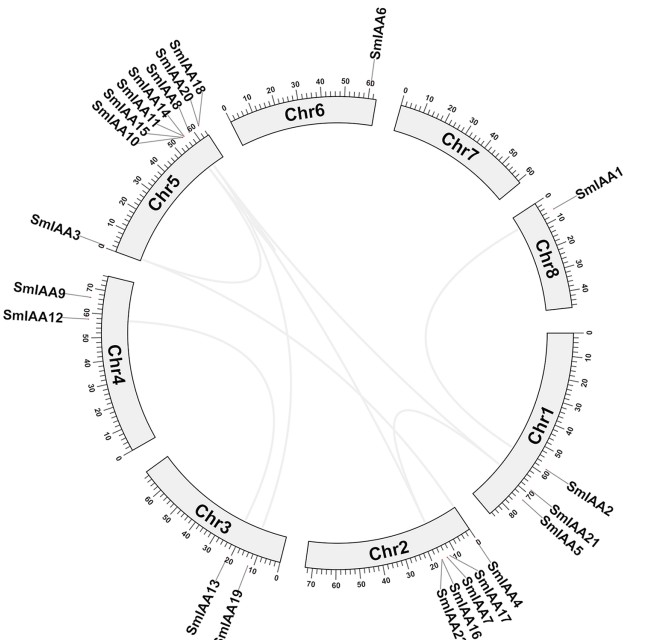

**Figure 4 Collinearity and selection pressure analysis of *SmIAAs*.** The collinearity analysis of *SmIAAs* was performed using the MCScanX software, and the results are shown as syntenic blocks. The lines connecting the genes represent the syntenic relationships between them. The collinearity analysis revealed several syntenic gene pairs, indicating the occurrence of gene duplication events during the evolution of *SmIAAs*. The Ka/Ks ratio analysis was used to measure the selection pressure acting on each syntenic gene pair. The Ka/Ks ratios were calculated using the KaKs_Calculator software.

and *SmIAA8/SmIAA19*, the Ka/Ks value of the other *S. miltiorrhiza* lineal homologous gene pairs is less than 1, which means that these *SmIAAs* evolved under purification selection, indicating that the sequence of the Aux/IAA gene is relatively conservative. These findings suggest that the Aux/IAA gene plays a significant role in *S. miltiorrhiza* and attains an optimal state, which is consistent with the conservative function of several *SmIAAs*.

## Analysis of GO enrichment and *cis*-acting element

The GO categorization system is a widely used method for characterizing genes and proteins in various organisms. The system is organized into three distinct categories, including molecular function, biological process, and cellular component, with each category corresponding to specific functions or properties. To investigate the potential functions of the *SmIAAs*, we utilized TBtools to examine the GO enrichment of *SmIAA* and identified the top 10 most significantly enriched GO terms across the three categories (if less than 10, all).

Figure 5A demonstrates that the *SmIAAs* are associated with 10, five, and three GO terms related to biological process, cellular component, and molecular function, respectively. The majority of *SmIAAs* are believed to play a role in various biological activities. Their molecular functions include transcription regulator activity, DNA-binding transcription factor activity, and identical protein binding. The cell composition mainly

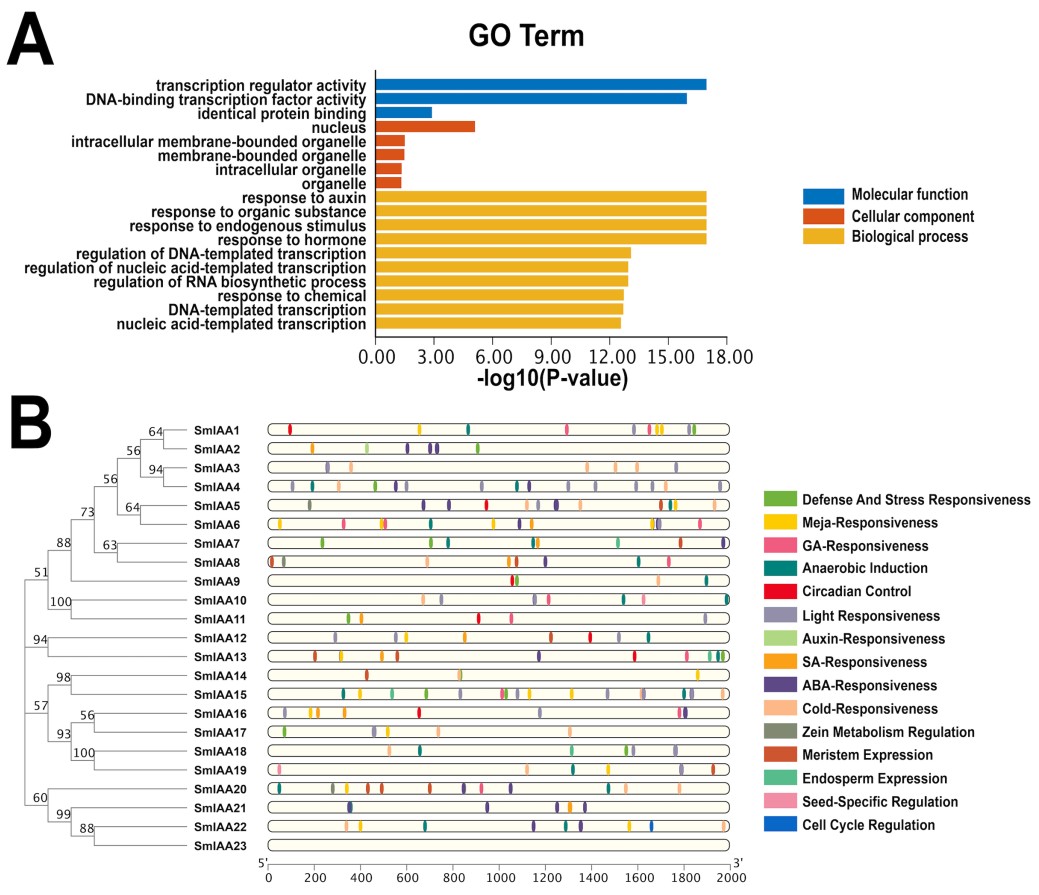

**Figure 5** **GO enrichment and various cis-element analysis of *SmIAAs*.** (A) GO enrichment analysis was performed to investigate the biological functions of the *SmIAAs*. The GO terms are grouped into three main categories: biological process, cellular component, and molecular function. (B) The presence of various *cis*-elements in the promoter regions of the SmIAA genes was analyzed using the PlantCARE database. The different colored bars represent different *cis*-elements. The *cis*-elements are grouped into different categories, including hormone-related, stress-related, and development-related *cis*-elements.

includes the nucleus, nuclear, intracellular membrane-bounded organelle, membrane-bounded organelle, and other cellular components. Furthermore, *SmIAAs* are primarily involved in biological processes related to the response to auxin, organic substance, endogenous stimulus, and hormone, among others.

To gain a more comprehensive understanding of the role of *SmIAAs* in the growth and development of *S. miltiorrhiza*, we analyzed the cis-acting elements in the promoter region of the related genes, which is 2,000 bp upstream of the initiation codon (Fig. 5B and Table S1). Our analysis revealed the presence of numerous cis-acting elements associated with auxin, salicylic acid, abscisic acid, methyl jasmonate, and gibberellin in the promoter region of *SmIAAs*, in addition to key cis-acting elements associated with transcription and light response. The presence of these cis-regulatory components suggests that hormones have the capacity to regulate *SmIAAs*. Overall, our analysis of cis-acting elements reveals that different family members contain a variety of response motifs, implying that the SmIAA gene family may participate in various physiological processes of *S. miltiorrhiza* by

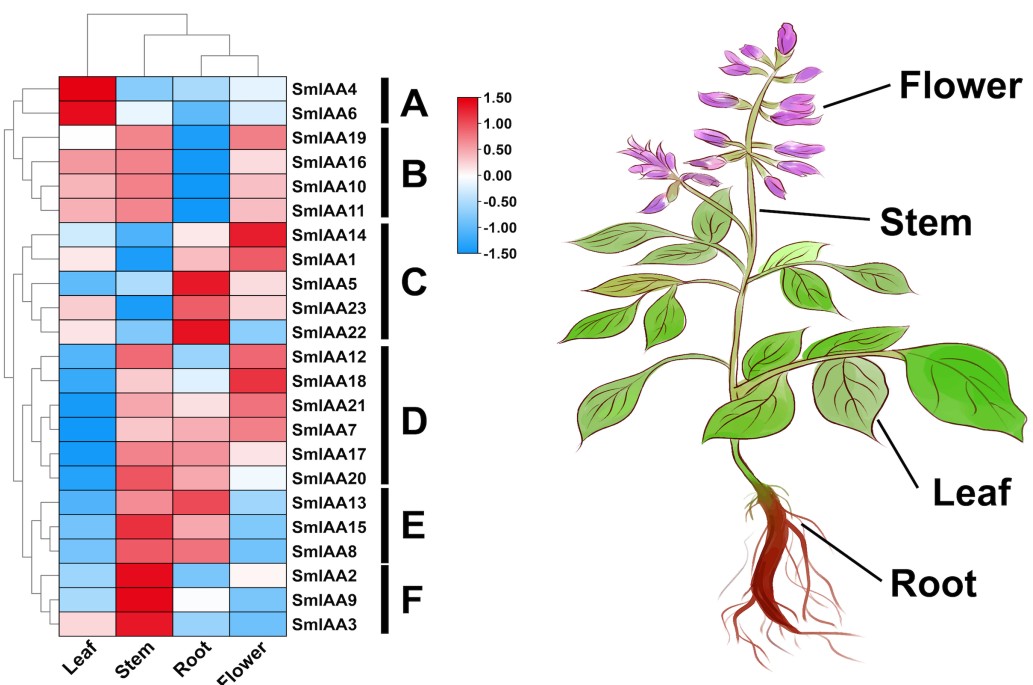

**Figure 6 (A–F) Expression profiles of *SmIAAs* in different tissues of *S. miltiorrhiza*.** The expression patterns of *SmIAAs* in different tissues of *S. miltiorrhiza* were analyzed using transcriptome data. Heat map visualization of the expression profiles was generated using TPM values, and row clustering was performed to group genes with similar expression patterns together. The heat map shows the expression levels of the *SmIAAs* across different tissues of *S. miltiorrhiza*, including root, stem, leaf, and flower. The color scale represents the relative expression level of each gene, with red indicating high expression and blue indicating low expression. The clustering of the genes reveals several groups with similar expression patterns, indicating potential functional similarities among these genes.

responding to various response elements such as hormone, low temperature, light, and so on.

## Analysis of gene expression pattern

The temporal and spatial expression of a gene may provide insights into its functional features (*Wang et al., 2021*). In this study, transcriptome data retrieved from NCBI was used to extract the TPM values of root, stem, leaf, and flower tissues of *S. miltiorrhiza*, and subsequently analyze the expression pattern of the *SmIAAs*. The results of gene expression patterns based on transcriptome data indicated considerable variability in the expression of several *SmIAAs* across different tissues of *S. miltiorrhiza*, as well as in the expression trends of individual genes. Based on the results of expression cluster analysis, *SmIAAs* could be categorized into six groups (A~F) (Fig. 6). Group A consisted of two genes (*SmIAA4* and *SmIAA6*) that were primarily expressed in the leaves of *S. miltiorrhiza*. Group B included four genes (*SmIAA19, SmIAA16, SmIAA10,* and *SmIAA11*) that showed similar expression levels in the leaves, stems, and flowers of *S. miltiorrhiza*, and relatively low expression levels in the roots. Group C comprised five genes (*SmIAA14, SmIAA1, SmIAA5, SmIAA23,* and *SmIAA22*) that were predominantly expressed in the flowers and

**Table 2 Different transcription factors in the *S. miltiorrhiza* genome.** The different transcription factors identified in the genome of *S. miltiorrhiza* using iTAK tool.

| Total | | | | 1814 | | | |
|---|---|---|---|---|---|---|---|
| Class | Num | Class | Num | Class | Num | Class | Num |
| Alfin-like | 7 | CAMTA | 4 | HB-other | 12 | OFP | 25 |
| AP2/ERF-AP2 | 19 | CPP | 8 | HB-PHD | 2 | PLATZ | 12 |
| AP2/ERF-ERF | 143 | CSD | 4 | HB-WOX | 15 | RWP-RK | 17 |
| AP2/ERF-RAV | 3 | DBB | 5 | HRT | 1 | S1Fa-like | 2 |
| B3 | 44 | DBP | 2 | HSF | 34 | SAP | 2 |
| B3-ARF | 26 | DDT | 8 | LFY | 1 | SBP | 17 |
| BBR-BPC | 6 | E2F-DP | 8 | LIM | 10 | SRS | 9 |
| BES1 | 9 | EIL | 7 | LOB | 50 | STAT | 1 |
| bHLH | 134 | FAR1 | 43 | MADS-M-type | 25 | TCP | 29 |
| bZIP | 67 | GARP-ARR-B | 12 | MADS-MIKC | 46 | Tify | 11 |
| C2C2-CO-like | 14 | GARP-G2-like | 48 | MYB | 145 | Trihelix | 31 |
| C2C2-Dof | 34 | GeBP | 10 | MYB-related | 65 | TUB | 12 |
| C2C2-GATA | 28 | GRAS | 62 | NAC | 84 | ULT | 1 |
| C2C2-LSD | 5 | GRF | 12 | NF-X1 | 3 | VOZ | 2 |
| C2C2-YABBY | 9 | HB-BELL | 15 | NF-YA | 8 | Whirly | 2 |
| C2H2 | 110 | HB-HD-ZIP | 43 | NF-YB | 21 | WRKY | 78 |
| C3H | 56 | HB-KNOX | 11 | NF-YC | 9 | zf-HD | 16 |

roots of *S. miltiorrhiza*. Group D encompassed six genes (*SmIAA12, SmIAA18, SmIAA21, SmIAA7, SmIAA17,* and *SmIAA20*) that showed the highest expression levels in the stems, roots, and flowers of *S. miltiorrhiza*, but minimal expression in the leaves. Group E consisted of three genes (*SmIAA13, SmIAA15,* and *SmIAA8*) that were primarily expressed in the stems and roots of *S. miltiorrhiza*. However, the three genes in group E (*SmIAA2, SmIAA9,* and *SmIAA3*) were significantly expressed only in the stem of *S. miltiorrhiza*. In summary, *SmIAAs* were expressed in clusters in various tissues of *S. miltiorrhiza*, with the majority of *SmIAAs* exhibiting high expression levels in roots, stems, and particularly flowers.

## Weighted correlation network analysis

Although the Aux/IAA gene family is crucial for auxin signal transduction, prior studies have suggested that its members may interact with other transcription factors (TFs) besides *ARFs* (*Luo, Zhou & Zhang, 2018*). To explore these interactions, we employed the iTAK tool to identify all TFs in the *S. miltiorrhiza* genome and utilized weighted correlation network analysis (WGCNA) to examine additional transcriptome data from *S. miltiorrhiza* flower tissue at different stages (withering, full blooming, and bud). Our analysis identified 68 TFs, including *ARF, ERF, MYB*, and others (Tables 2 and S6). We used a soft threshold of β = 14 to divide the 1814 TFs into 10 modules, with gene numbers ranging from 45 to 490 in each module (Table S7). The outcomes of the GO

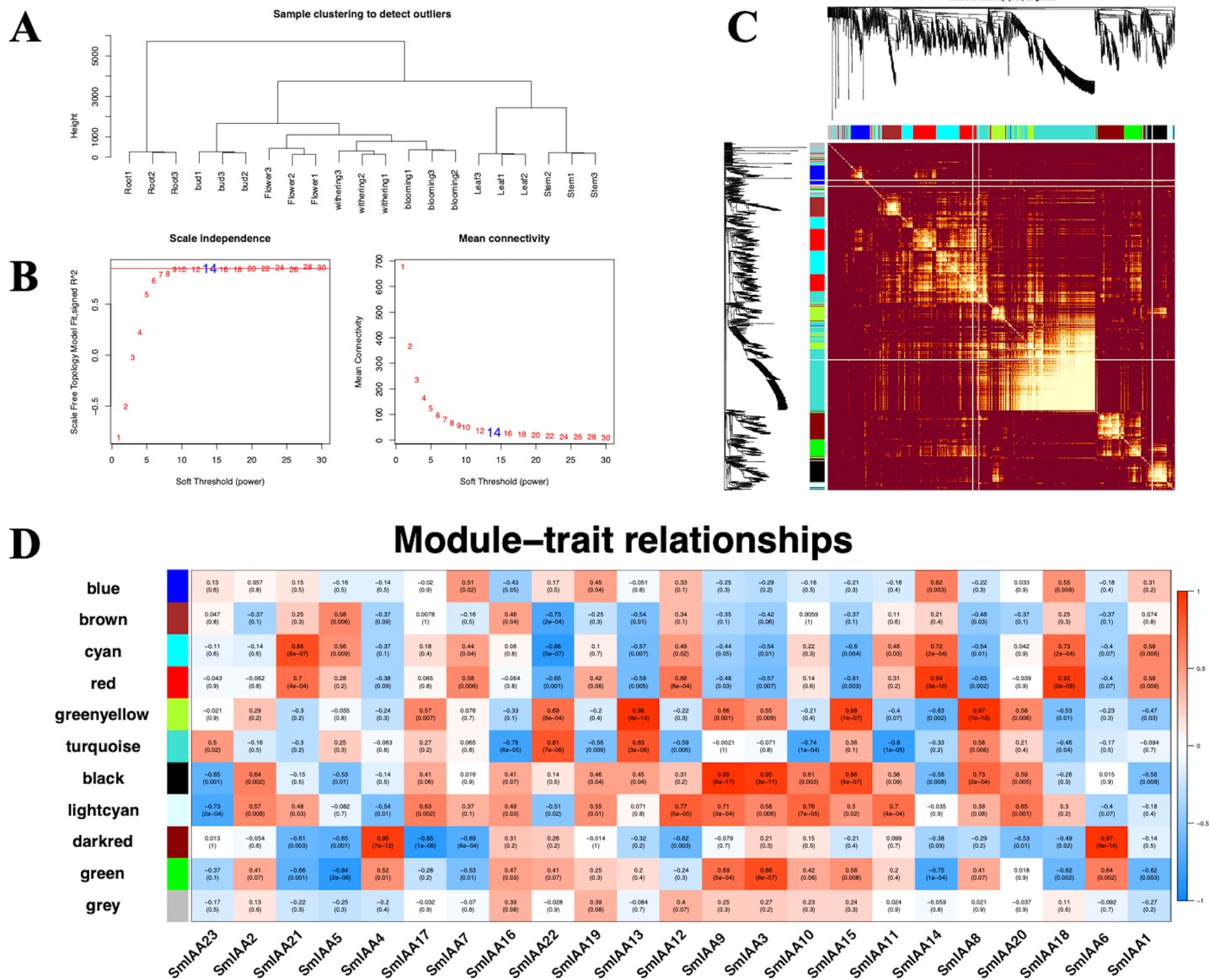

**Figure 7 (A–D) The correlation diagram between the module and *SmIAAs* calculated by the WGCNA method.** To investigate the correlation between SmIAA genes and different gene modules, a weighted gene co-expression network analysis (WGCNA) was performed based on transcriptome data from different tissues of *S. miltiorrhiza*.

enrichment analysis of distinct modules revealed that each of these modules has distinct biological significance (Table S8). Furthermore, utilizing the Pearson correlation data, we observed a significant association between *SmIAAs* and the modules, suggesting that *SmIAAs* may be linked with the module hub genes (Fig. 7). We selected the red and cyan modules for further investigation, as they were highly associated with *SmIAA14* and negatively related to *SmIAA22*, respectively. We identified the top 10 hub genes in each module and annotated their homologs in the iTAK database (Table 3). Surprisingly, only one *ARF* was among the 20 key genes identified across various modules (EVM0019866),
**Table 3 Top 10 hub genes in the black and cyan modules.** The hub genes were identified using the weighted gene co-expression network analysis (WGCNA) method. The table includes information about the module color, gene name, gene annotation, correlation, *p*-value, and the K-MEANS-based connectivity (KME).

| Group | Gene | Annotation | R | P | KME |
|-------|------|-----------|------|------|------|
| Black | EVM0007532 | WRKY | 0.9931 | 3.72E−19 | 0.9931 |
| Black | EVM0000041 | bHLH | 0.9929 | 5.06E−19 | 0.9929 |
| Black | EVM0019585 | zf-HD | 0.9865 | 2.14E−16 | 0.9865 |
| Black | EVM0012769 | MYB | 0.9857 | 3.78E−16 | 0.9857 |
| Black | EVM0010344 | bHLH | 0.9851 | 5.29E−16 | 0.9851 |
| Black | EVM0018729 | HB-WOX | 0.9840 | 1.05E−15 | 0.9840 |
| Black | EVM0021887 | AP2/ERF-ERF | 0.9800 | 8.72E−15 | 0.9800 |
| Black | EVM0023258 | SBP | 0.9797 | 9.96E−15 | 0.9797 |
| Black | EVM0015830 | zf-HD | 0.9791 | 1.32E−14 | 0.9791 |
| Black | EVM0011877 | bHLH | 0.9737 | 1.15E−13 | 0.9737 |
| Cyan | EVM0017797 | DBB | 0.9906 | 7.35E−18 | 0.9906 |
| Cyan | EVM0019866 | B3-ARF | 0.9763 | 4.30E−14 | 0.9763 |
| Cyan | EVM0001762 | C3H | 0.9744 | 9.03E−14 | 0.9744 |
| Cyan | EVM0004701 | MYB | 0.9731 | 1.45E−13 | 0.9731 |
| Cyan | EVM0002416 | HSF | 0.9697 | 4.34E−13 | 0.9697 |
| Cyan | EVM0011962 | bZIP | 0.9662 | 1.23E−12 | 0.9662 |
| Cyan | EVM0004337 | MYB-related | 0.9645 | 1.91E−12 | 0.9645 |
| Cyan | EVM0016656 | C2H2 | 0.9607 | 5.02E−12 | 0.9607 |
| Cyan | EVM0025053 | MYB-related | 0.9584 | 8.45E−12 | 0.9584 |
| Cyan | EVM0000238 | C2C2-Dof | 0.9583 | 8.73E−12 | 0.9583 |

suggesting that *SmIAAs* may have other important functions beyond their role in the auxin pathway. However, more research is needed to confirm this hypothesis.

### The validation of qRT-PCR

To confirm the RNA-Seq results, we employed the quantitative reverse transcription polymerase chain reaction (qRT-PCR) technique to evaluate the expression levels of 12 genes, comprising of four *SmIAAs*, four hub genes selected by co-expression modules, and four transcription factors chosen randomly (Fig. 8). The findings demonstrated that the alteration in gene expression detected by qRT-PCR and RNA-Seq techniques was similar, and the outcomes of the two datasets were positively correlated, suggesting that the co-expression network established in this study is reliable.

## DISCUSSION

The Aux/IAA gene family is a large and diverse family of transcriptional regulators that plays a critical role in the response pathway of plant auxin. The family has been identified and studied in a wide range of plant species, from soybean to wheat, tomato, and March plum (*Gao et al., 2016*; *Jain et al., 2006*; *Liscum & Reed, 2002*; *Luo, Zhou & Zhang, 2018*).

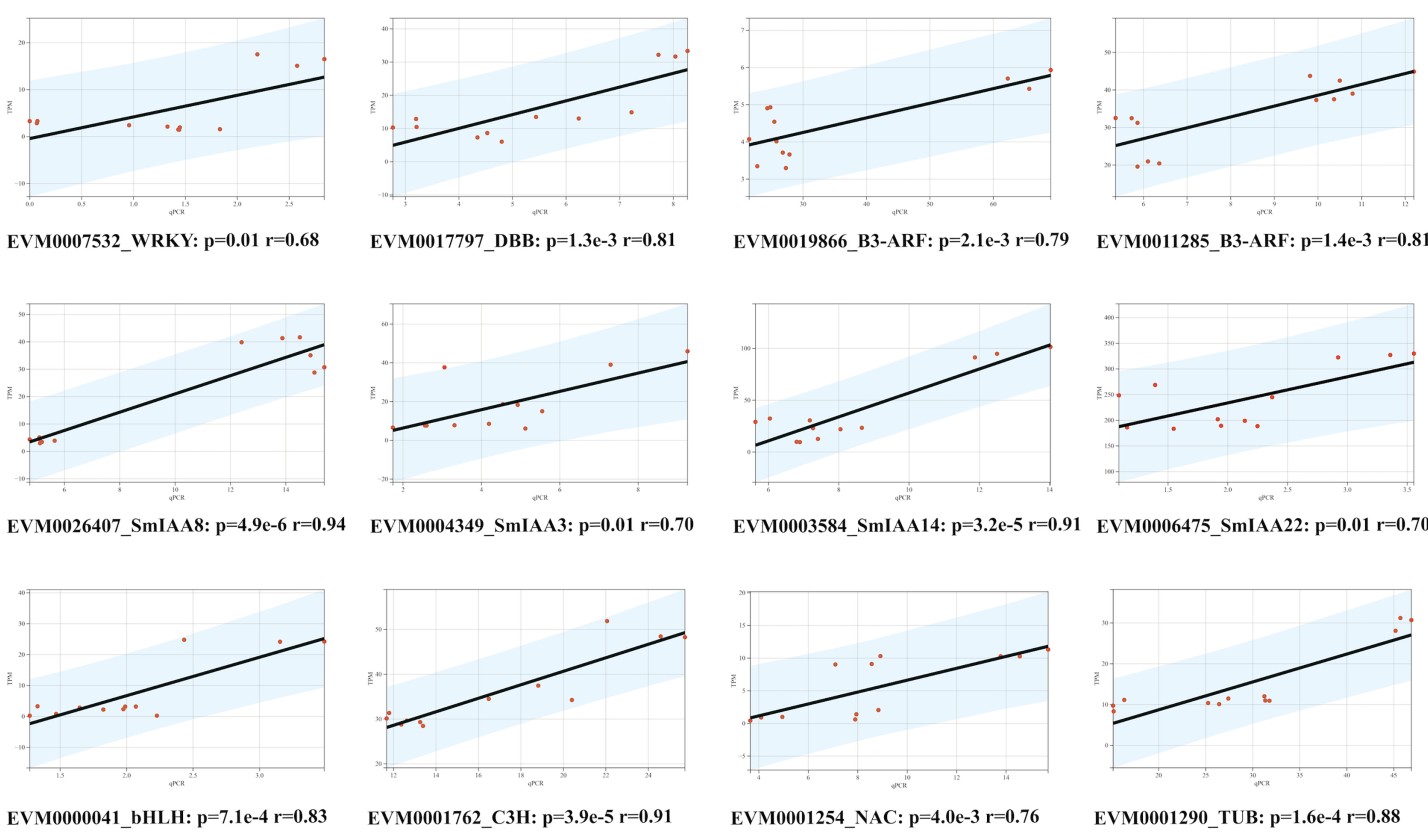

**Figure 8 Correlation analysis between RNA-seq and qPCR data of 12 randomly selected *SmIAAs*.** To validate the RNA-seq data, quantitative PCR (qPCR) was performed on 12 randomly selected *SmIAAs*. The expression levels of these genes were quantified by both RNA-seq and qPCR, and the correlation coefficient ($R^2$) was calculated to compare the results. The results presented in this figure demonstrate a strong positive correlation ($R^2 > 0.8$) between the RNA-seq and qPCR data for all 12 genes, indicating that the RNA-seq data is reliable and accurate.

The family members share a common structure that includes four conserved domains, and their expression is regulated by the presence and concentration of auxin in plant cells. This study focuses on the identification, structural, and phylogenetic analysis of the Aux/IAA gene family in *S. miltiorrhiza*. On the whole, the number of *SmIAAs* is similar to that of Aux/IAA gene family members in *A. thaliana* (*Liscum & Reed, 2002*). Most of the *SmIAAs* are located on chromosomes other than chromosome 7, and many of them are situated at the ends of the chromosomes. This suggests that the expansion of the *SmIAAs* may have been caused by tandem repeats.

Protein domains are important functional units, and conserved motifs are subunits of domains that contribute to the diverse biological functions of the domains. In the proteins encoded by the *SmIAAs*, 17 members contained four typical conserved domains of the family, while the remaining six members had deletions of these typical domains. These atypical Aux/IAA proteins are mainly missing Domain I (EAR-motif), so it is inferred that the transcriptional inhibitory function of these six members of *S. miltiorrhiza* on downstream ARF may be missing (*Luo, Zhou & Zhang, 2018*). It is reported that atypical Aux/IAA proteins are ubiquitous in various plant *Aux/IAA* gene families and play an

**Table 4 The relationship between different modules and different tissues of _S. miltiorrhiza_.** The modules were identified using the weighted gene co-expression network analysis (WGCNA) method. The table includes information about the module color, _SmIAAs_ in each module, and the tissues in which the module was highly expressed.

| Module | R | Gene | Tissue |
| --- | --- | --- | --- |
| Green | −0.62 | SmIAA1 | Flower |
| Darkred | −0.69 | SmIAA7 | Flower |
| Lightcyan | 0.77 | SmIAA12 | Flower |
| Red | 0.94 | SmIAA14 | Flower |
| Red | 0.92 | SmIAA18 | Flower |
| Cyan | 0.86 | SmIAA21 | Flower |
| Darkred | 0.96 | SmIAA4 | Leaf |
| Darkred | 0.97 | SmIAA6 | Leaf |
| Green | −0.84 | SmIAA5 | Root |
| Cyan | −0.86 | SmIAA22 | Root |
| Lightcyan | −0.73 | SmIAA23 | Root |
| Black | 0.64 | SmIAA2 | Stem |
| Black | 0.95 | SmIAA3 | Stem |
| Black | 0.99 | SmIAA9 | Stem |
| Lightcyan | 0.76 | SmIAA10 | Stem&Flower |
| Turquoise | −0.8 | SmIAA11 | Stem&Flower |
| Turquoise | −0.76 | SmIAA16 | Stem&Flower |
| Turquoise | −0.56 | SmIAA19 | Stem&Flower |
| Greenyellow | 0.97 | SmIAA8 | Stem&Root |
| Greenyellow | 0.98 | SmIAA13 | Stem&Root |
| Greenyellow | 0.88 | SmIAA15 | Stem&Root |
| Darkred | −0.85 | SmIAA17 | Stem&Root |
| Lightcyan | 0.65 | SmIAA20 | Stem&Root |

important role in plant adaptation to changing environmental conditions, but more specific functions still need to be further studied (_Li et al., 2017_; _Luo, Zhou & Zhang, 2018_).

In addition to four typical conserved motifs, there are six conserved motifs in Aux/IAA protein. The overlap and difference of conserved motifs among different members reflect its functional characteristics to some extent. Other different conserved motifs may enable members to participate in functions other than auxin signal pathway, and the different expression patterns of members in different tissues of _S. miltiorrhiza_ indicate that these different conserved motifs play different roles to a certain extent, but the specific functions of their effects need to be further studied. Moreover, the promoter analysis of _SmIAAs_ suggests that they may respond to various stimuli, including auxin, gibberellin, abscisic acid, salicylic acid, methyl jasmonate, drought, salt, heat stress, and low temperature. The presence of these cis-acting elements may explain why _SmIAAs_ have a wide range of functions in _S. miltiorrhiza_, including tissue-specific roles. However, further experimental research is required to confirm this. Overall, the physicochemical properties of SmIAA

proteins contribute to their regulation of gene expression and involvement in plant adaptation to changing environmental conditions.

Transcriptome analysis showed that the expression of different Aux/IAA family genes was tissue specific, but Aux/IAA family genes were involved in each different tissue of *S. miltiorrhiza*. Previous reports have pointed out that *Aux/IAA* gene not only interacts with ARF, but also is regulated by other transcription factors (*Guilfoyle & Hagen, 2007*; *Luo, Zhou & Zhang, 2018*). In this study, co-expression network analysis was performed using transcriptome data from *S. miltiorrhiza* to identify transcription factors that may interact with the *SmIAAs*. This analysis identified 10 modules, and further correlation analysis revealed that each module was negatively correlated with at least one *SmIAA*. Analysis of the specific expression patterns of different *SmIAAs* in different tissues suggests that the different modules may play different roles in different tissues of *S. miltiorrhiza* (Table 4). Further analysis of the black and red modules revealed an interesting finding: although the Aux/IAA family is known to play an important role in auxin signaling, the hub genes in these two modules had only a small number of *ARFs* that were significantly associated with *SmIAAs*. This suggests that the *SmIAAs* may have additional, unknown functions in *S. miltiorrhiza*, but further research is needed to confirm this.

## CONCLUSIONS

In this study, a comprehensive analysis was conducted on the Aux/IAA gene family in *S. miltiorrhiza*, which included whole genome identification, physicochemical property analysis, and expression analysis. Moreover, a gene co-expression network was constructed to investigate the interactions of *SmIAAs* with transcription factors. The findings revealed that this gene family comprised numerous members with diverse structures and functions. Notably, it was observed that these genes are strongly correlated with transcription factors other than ARF. For instance, *SmIAA9* was found to cooperate with the *WRKY* (EVM0007532) to regulate stem growth, while *SmIAA22* and the *DBB* (EVM0017797) were observed to antagonize each other and regulate root development. These results suggest that *SmIAAs* perform various unknown functions in different tissues and during different growth and development processes of *S. miltiorrhiza*. The outcomes of this research provide a foundation for further elucidation of the function of the *SmIAAs* and offer valuable resources for improving the characteristics of *S. miltiorrhiza* and breeding new *S. miltiorrhiza* varieties through genetic engineering.

### Funding

This research was supported by the Hunan Provincial Natural Science Foundation of China (2021JJ40386) and the Scientific Research Incubation Fond of Hunan University of Medicine (No. 20KJPY07). The funders had no role in study design, data collection and analysis, decision to publish, or preparation of the manuscript.

## Grant Disclosures

The following grant information was disclosed by the authors:

Hunan Provincial Natural Science Foundation of China: 2021JJ40386.

Scientific Research Incubation Fond of Hunan University of Medicine: 20KJPY07.

## Competing Interests

The authors declare that they have no competing interests.

## Author Contributions

- Bin Huang conceived and designed the experiments, performed the experiments, analyzed the data, prepared figures and/or tables, authored or reviewed drafts of the article, and approved the final draft.
- Yuxin Qi performed the experiments, prepared figures and/or tables, and approved the final draft.
- Xueshuang Huang conceived and designed the experiments, authored or reviewed drafts of the article, and approved the final draft.
- Peng Yang conceived and designed the experiments, authored or reviewed drafts of the article, and approved the final draft.

## Data Availability

The original q-PCR data is available in the Supplemental File.

## Supplemental Information

Supplemental information for this article can be found online at http://dx.doi.org/10.7717/peerj.15212#supplemental-information.

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
