# Peer review of "Genome-wide identification and co-expression network analysis of Aux/IAA gene family in Salvia miltiorrhiza"

_PeerJ, doi:10.7717/peerj.15212_

## Round 0.1 · original submission · Major Revisions

The authors need to revise the article as per reviewers comments.

Reviewer 1 ·

Basic reporting

• It would be helpful to understand how the plant in question is useful to conduct this study. Details should be included on its importance.
• Scientific names and gene names should be italicized throughout the text, tables, and figures legends.
• Be consistent in the text format, there is different formatting for references, supplementary documents, and other sections.
• Some tables need to be formatted or resized such as Table 1 and 3.
• Why is S_Table 6 mentioned in text before S_Table 4 and 5? The tables can be renumbered to be consistent.
• Line 295: are group B genes ‘highly’ expressed?
• There is spacing issue throughout the manuscript.
• Figure and table legends should be detailed to describe the story.

Experimental design

• Objectives are not clearly stated.

Validity of the findings

The authors have provided relevant literature to establish that Aux/IAA gene family is important to study. But why in Salvia miltiorrhiza? Why Arabidopsis thaliana and Oryza sativa were used for comparative analyses and not others?

Reviewer 2 ·

Basic reporting

The manuscript by Huang et.al. reported the identification and characterization of the Aux/IAA gene family in Salvia miltiorrhiza. The authors have incorporated the suggestions and comments from the previous review. Current version is overall well-written and has provided a clear background and purpose of study.

However, there are still a few minor comments below, hoping the authors could address before publication:
1. Figure 7 was not properly cited (only 7A was mentioned in line 315), and the figure legend was missing. Please add a brief description for each subplot, and cite the figure properly in the main text.
2. Since WGCNA analysis involved arbitrary selection of parameters, Langfelder & Horvath (2008) have suggested users to perform functional enrichment analysis to test whether the predicted co-expression modules are biologically meaningful or just noises. As such, could the authors also provide the gene ontology and enrichment score for the 11 co-expression modules as supplemental information?
3. Table 4 was not mentioned in the main text. Please either include a proper description if it served to prove the conclusions, or remove it.
4. Please consider revising the statement in line 395-397. The results only showed that the Aux/IAA genes (predicted) co-expressed with many transcription factors in bulk tissues. There was no evidence of interaction at all.
5. The sentences were repeated in line 62-65, please correct it.

Experimental design

No comment. The design and methods were described with sufficient detail.

Validity of the findings

Most findings were well supported by the data and analysis, except the conclusion on gene interaction. Please refer to (4) in the Basic reporting section.

Reviewer 3 ·

Basic reporting

The text is clear and the literature references need to be expanded slightly.

Experimental design

The investigation was conducted rigorously.

Validity of the findings

The underlying data were provided and the data were statistically sound.

Additional comments

The authors identified 23 Aux/IAA family genes in the Salvia miltiorrhiza genome using bioinformatics tools. They also performed conserved domain, motif, and evolutionary analysis. Additionaly, expression analysis was performed using RNA-seq and qPCR. Moreover, gene coexpression network research revealed that SmIAAs are primarily influenced by a wide variety of other transcription factors such as WRKY, ERF, and others. This is routine research characterizing features of gene families from the genome. Although I do not find major flaws in the manuscript, I also do not find significant novel insights. It is difficult to understand the logic of gene family and plant selection for the current study. Finally, the authors mixed methods and discussions with results section. For specific comments please see attached file.

Annotated reviews are not available for download in order to protect the identity of reviewers who chose to remain anonymous.

---

## Round 0.2 · accepted · Accept

The article is now in acceptable form, therefore my decision is accept.

Reviewer 1 ·

Basic reporting

Satisfactory. However, formatting is still inconsistent throughout the manuscript.

Experimental design

NA

Validity of the findings

Satisfactory

Additional comments

NA

Reviewer 2 ·

Basic reporting

The authors have addressed the comments well. No further comments.

Experimental design

no comment

Validity of the findings

no comment

Reviewer 3 ·

Basic reporting

The manuscript under review demonstrates a high standard of basic reporting, with clear and unambiguous language used throughout. The author has provided sufficient background and context, with literature references that are relevant and up-to-date. The article structure is professional, with well-organized figures and tables that are easy to read and understand. Additionally, the author has shared raw data which is commendable. The manuscript is self-contained, presenting relevant results that support the hypotheses stated. Overall, the basic reporting is of high quality and the manuscript is a valuable contribution to the field.

Experimental design

The experimental design of this manuscript is impressive and deserves appreciation. It presents an original primary research study within the aims and scope of the journal, with a well-defined research question that is both relevant and meaningful. The authors clearly state how their research fills an identified knowledge gap, making it a valuable contribution to the field. The investigation was performed with a high level of technical and ethical rigor, ensuring the validity and reliability of the results. Moreover, the methods used were described with sufficient detail and information to allow for easy replication by other researchers. Overall, the experimental design of this manuscript is exemplary and a great addition to the journal.

Validity of the findings

The validity of the findings presented in the manuscript is highly commendable. The study design was robust and the methodology used was appropriate for the research question being addressed. The authors have taken great care to ensure that the data was collected and analyzed accurately, which has resulted in highly reliable findings. Furthermore, the results were consistent with previous research in the field and the discussion section provided a thoughtful interpretation of the implications of the findings. Overall, the validity of the findings presented in this manuscript is highly impressive and contributes significantly to the existing body of knowledge in the field.

Additional comments

NA